# A user-friendly, open-source tool to project impact and cost of diagnostic tests for tuberculosis

David W Dowdy[1,2]*, Jason R Andrews[3†], Peter J Dodd[4], Robert H Gilman[5]

[1]Department of Epidemiology, Johns Hopkins Bloomberg School of Public Health, Baltimore, United States; [2]Center for Tuberculosis Research, Johns Hopkins University, Baltimore, United States; [3]Division of Infectious Diseases, Department of Medicine, Massachusetts General Hospital, Boston, United States; [4]School of Health and Related Research, University of Sheffield, Sheffield, United Kingdom; [5]Department of International Health, Johns Hopkins Bloomberg School of Public Health, Baltimore, United States

**Abstract** Most models of infectious diseases, including tuberculosis (TB), do not provide results customized to local conditions. We created a dynamic transmission model to project TB incidence, TB mortality, multidrug-resistant (MDR) TB prevalence, and incremental costs over 5 years after scale-up of nine alternative diagnostic strategies. A corresponding web-based interface allows users to specify local costs and epidemiology. In settings with little capacity for up-front investment, same-day microscopy had the greatest impact on TB incidence and became cost-saving within 5 years if delivered at $10/test. With greater initial investment, population-level scale-up of Xpert MTB/RIF or microcolony-based culture often averted 10 times more TB cases than narrowly-targeted strategies, at minimal incremental long-term cost. Xpert for smear-positive TB had reasonable impact on MDR-TB incidence, but at substantial price and little impact on overall TB incidence and mortality. This user-friendly modeling framework improves decision-makers' ability to evaluate the local impact of TB diagnostic strategies.

*For correspondence: ddowdy1@jhmi.edu

Present address: †Division of Infectious Diseases, Department of Medicine, Stanford University School of Medicine, Palo Alto, United States

Competing interests: The authors declare that no competing interests exist.

## Introduction

Infectious disease transmission models are important tools for translating the best current knowledge of the natural history and epidemiology of infectious diseases into projections of epidemiological impact (e.g., incidence, mortality) and costs under alternative strategies for disease control (*Garnett et al., 2011*). Currently, most published transmission models are either loosely calibrated to reflect global/regional outcomes or more tightly fit to specific epidemiological settings; in either case, model results may be difficult for local decision-makers in the majority of public health settings to utilize. Simplified models designed for in-country use by decision-makers, most notably the Spectrum suite of models supported by the Futures Institute (*Stover et al., 2010*), have been used to inform decision-making in the fields of reproductive health and human immunodeficiency virus (HIV) for over a decade (*Stover, 2004*). Estimates from the Spectrum models are now routinely incorporated into official global and country-level estimates of HIV disease burden (*Brown et al., 2010*) and intervention impact (*Farnham et al., 2013*). Other simplified models are readily available for impact projections related to non-infectious diseases, where transmission assumptions are less important (*Betz Brown et al., 2000*; *Walker et al., 2013*). However, to date, simple, user-friendly transmission models have not been widely used for decision-making related to many infectious diseases other than HIV. Diagnosis of active tuberculosis (TB) is an example of a public health intervention for which transmission models

**eLife digest** Tuberculosis is an infectious bacterial disease caused predominantly by the microorganism *Mycobacterium tuberculosis*. Although the number of deaths from tuberculosis has been falling in recent years, the disease still kills more than 1 million people every year, mainly in developing countries. Tuberculosis can be treated with antibiotics, but the emergence of bacteria that are resistant to existing drugs is threatening efforts to eradicate the disease.

Preventing the spread of tuberculosis is heavily dependent on accurate diagnosis of individuals with the disease. This is challenging because the initial symptoms are often mild, usually just a cough, which means that someone can spread the disease to many others over a period of several months before the symptoms become worse—fever, night sweats, and weight loss—and they realize that they are sick. Multiple diagnostic strategies are available, from the relatively low-tech—examining sputum samples under a microscope to detect tuberculosis bacteria—to more sophisticated tests that can detect bacterial DNA and determine whether the bacteria are drug-resistant in less than 2 hr.

Choosing which diagnostic strategy to adopt can be challenging because the optimal solution in a region will depend on the specific local conditions. To overcome this problem, Dowdy et al. have developed a computer program that enables decision-makers to input four key parameters that describe the tuberculosis situation in their region, and to obtain 5-year projections of the rate of new infections, mortality, and total costs likely to result from adopting any of nine different diagnostic strategies.

The four parameters are the number of new cases of tuberculosis each year (incidence), the proportion of new cases that are multi-drug resistant, the proportion of the adult population that has HIV, and the local costs of various diagnostic techniques and treatments. Since the entire computer program is written in a freely available open-source programming language (Python), any user can tweak these parameters to provide a more precise fit to their own region. Alternatively, the standard version of the program can be run directly from a website without any need to interact with computer code.

This model is the first to enable local decision-makers to evaluate the impact of different diagnostic strategies for tuberculosis under the conditions specific to their region. The model predicts, for example, that in areas where there is little money available for up-front investment, same-day microscopy analysis of sputum samples and starting patients on treatment is the most cost-effective strategy for reducing the rate of new infections. Given the wide variation in conditions within even small geographical areas, this more flexible approach should lead to the more efficient use of resources and may, ultimately, help to reduce the spread of tuberculosis.

may provide guidance on both global (*Dowdy et al., 2006*; *Abu-Raddad et al., 2009*) and country-specific levels (*Menzies et al., 2012*). Specifically, an unprecedented number of new diagnostic strategies for active TB are now recommended by the World Health Organization (WHO), including same-day microscopy (*World Health Organization, 2011*), microcolony-based culture techniques (*Leung et al., 2012*) such as the microscopic-observation drug-susceptibility (MODS) assay (*Moore et al., 2006*), line-probe assays for drug susceptibility testing (*Bwanga et al., 2009*), and Xpert MTB/RIF ('Xpert'), a molecular assay capable of providing results (including rifampin resistance) in 90 min with minimal human resource requirements (*Boehme et al., 2010*, *2011*). TB program decision-makers must repeatedly determine when to invest in scaling up a novel diagnostic test, which test(s) to promote, and whether the implementation strategy should differ by epidemiological situation (*Cobelens et al., 2012*). Without transmission models to provide locally relevant estimates of cost and impact under alternative implementation strategies, such decisions will be made without systematically considering the implications of available scientific evidence.

To aid in this decision-making process, we created a flexible, simple modeling tool that allows non-expert users to define their local situation according to three key epidemiological parameters (TB incidence, proportion of new TB cases that are multidrug-resistant [MDR], and adult human immunodeficiency virus [HIV] prevalence) and local unit costs of TB diagnosis and treatment. This tool then

incorporates those estimates into a combined decision analysis-transmission framework to generate 5-year projections of TB incidence, mortality, and control costs for nine diagnostic strategies (*Figures 1 and 2*). These strategies are:

1. 'Baseline': Sputum smear microscopy for each diagnostic attempt, with liquid-media TB culture only to evaluate smear-positive cases with a history of previous TB treatment for drug resistance. (Cultures in all scenarios trigger drug-susceptibility testing if positive.)
2. 'TB culture if previously treated': Sputum smear microscopy used for patients without a history of TB treatment; smear plus liquid-media culture used to diagnose TB in any previously treated individual with symptoms (regardless of smear status).
3. 'Xpert if HIV-positive': Xpert MTB/RIF for HIV-infected patients only, with a positive test for rifampin resistance triggering treatment for MDR-TB. Xpert is assumed to be deployed at the district level, such that results cannot generally be provided during the same clinical encounter (*Lawn et al., 2012*). This strategy is conceived as a 'best-case' scenario for HIV-targeted TB testing: if individuals unaware of their HIV status are not tested with Xpert, this strategy will overestimate effectiveness, and if those unaware of their status are tested, it will underestimate costs.
4. 'Xpert if Smear-Positive': Xpert MTB/RIF for smear-positive patients only (i.e., for rapid DST), with a positive test for rifampin resistance triggering treatment for MDR-TB.
5. 'Xpert for All': Xpert MTB/RIF for all patients.
6. 'Xpert with Culture DST Confirmation': Xpert MTB/RIF for all patients, but treatment for MDR-TB only initiated if rifampin resistance is confirmed by culture.
7. 'MODS/TLA': Sputum smear, plus microcolony-based TB culture (e.g., MODS or thin-layer agar, TLA) for all patients.
8. 'Same-Day Microscopy': Double the per-test cost of sputum smear microscopy, in exchange for the ability to provide results to patients in the same clinical encounter (e.g., with peripheral, unbatched reading of sputum smears).
9. 'Same-Day Xpert': Double the per-test cost of Xpert MTB/RIF, in exchange for the ability to provide results to patients in the same clinical encounter (e.g., peripheral deployment, with greater costs reflecting lower volume per machine [*Vassall et al., 2011*]).

For purposes of illustration, we evaluated each of these diagnostic strategies in four emblematic epidemiological settings, defined by TB incidence, MDR-TB prevalence among new cases, and adult HIV prevalence:

1. 'Reference/High-Incidence Setting' (e.g., Southeast Asia): TB incidence 250 per 100,000/year (twice the global incidence [*World Health Organization, 2012*]), MDR-TB prevalence of 3.7% in new TB cases (the estimated global prevalence [*World Health Organization, 2012*]), adult HIV prevalence of 0·83% (the estimated global prevalence [*UNAIDS, 2012*]);
2. 'Low-Incidence Setting' (e.g., United States, Western Europe): TB incidence at entry of 8.9 per 100,000/year and declining, with similar MDR-TB (as a proportion of new cases) and HIV prevalence as above;
3. 'High MDR Setting' (e.g., former Soviet Union): TB incidence 100 per 100,000/year, MDR-TB prevalence of 10% in new TB cases, adult HIV prevalence of 0.83%; and
4. 'High HIV Setting' (e.g., sub-Saharan Africa): Adult HIV prevalence of 20%, TB incidence 500 per 100,000, and MDR-TB prevalence among new cases of 3.7%.

In each setting, we used a uniform set of costs for purposes of comparison (*Table 1*). Although these four settings form the basis of the results presented here, decision-makers can use the open-source model program (included as *Supplementary file 1*, written in the open-source programming language Python, Version 2.7, www.python.org) to re-define any parameter in *Table 1* according to their best local knowledge; a manual for doing so is also included as *Supplementary file 2*. Capacity to create non-equilibrium settings (e.g., declining TB incidence, increasing MDR-TB) is included. We also provide a web-based interface (flexdx.modeltb.org) that allows users to input local TB incidence, MDR-TB prevalence, HIV prevalence, and unit costs; this interface provides customized results without the requirement to manipulate programming code. The program corresponding to the web version is also available on a public repository (https://github.com/JJPennington/FlexDx-TB-Web-Django).

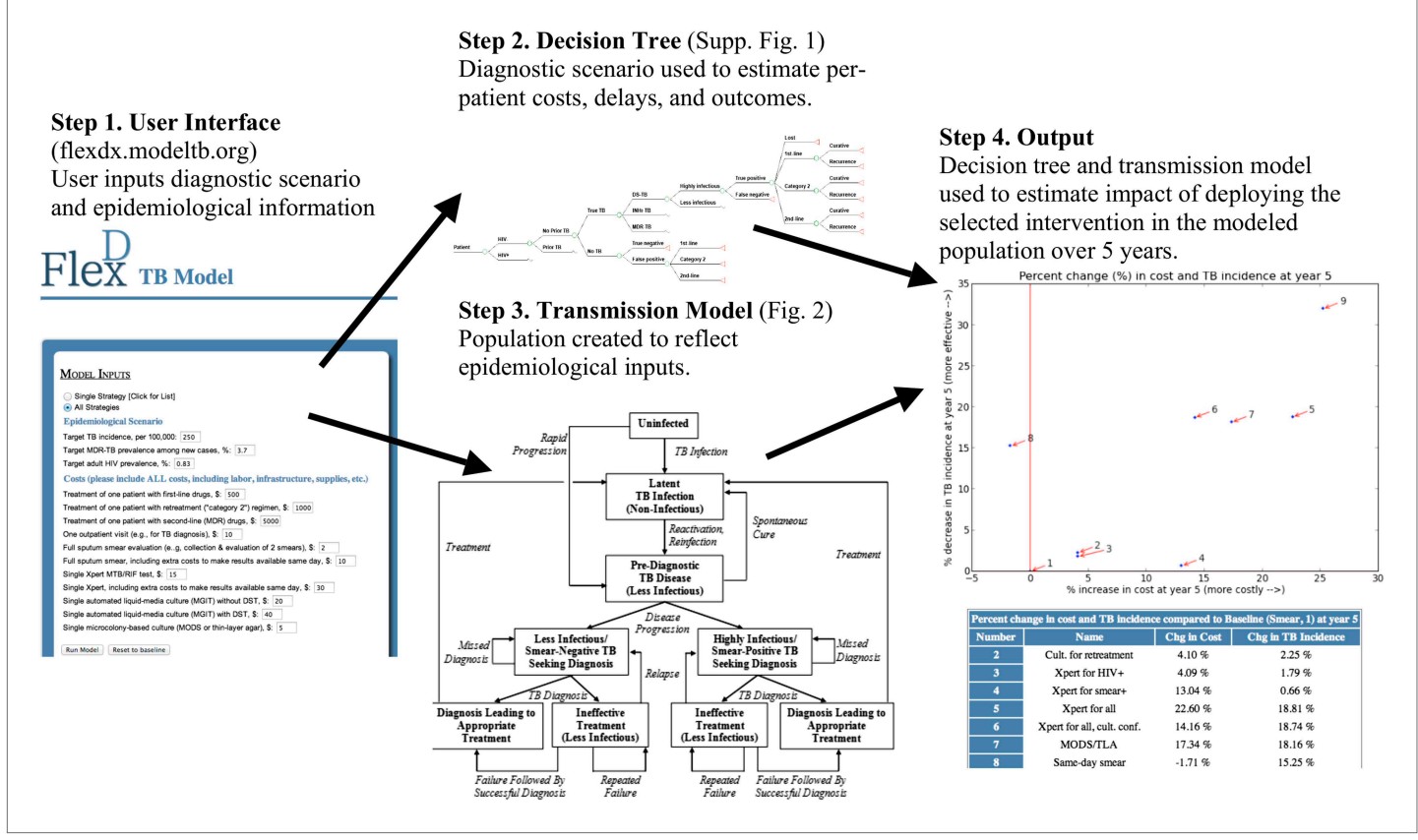

**Figure 1**. Overview of user-friendly model. Users are asked, via open-source computer script or Web interface, to select one of the nine diagnostic strategies and to provide unit costs and three basic epidemiological parameters (TB incidence, MDR-TB prevalence among new cases, and adult HIV prevalence). The selected diagnostic strategy is used to populate a decision tree that calculates (a) the probability of missed diagnosis, unsuccessful treatment, and successful treatment, (b) costs, and (c) diagnostic delays. These outputs depend on patients' TB (yes/no, and drug susceptibility status), HIV, and TB treatment history status. The selected epidemiological parameters are then used to populate a dynamic transmission model, creating a steady-state population that reflects local TB epidemiology. The decision tree—which inputs user-defined unit costs—is then incorporated into the transmission model to project outcomes under the selected diagnostic scenario. Users can sequentially select multiple diagnostic scenarios for comparison, and the computer script (though not the Web interface) allows users to manipulate input parameters at their discretion.

## Results

### Model validation

To validate the model, we compared selected model outcomes to published global estimates. In the high-incidence setting, our model estimated TB mortality at 14% of incidence (95% uncertainty range: 7–20%, WHO global estimate 14% [*World Health Organization, 2012*]), HIV-associated TB at 13% of all incident TB (95% uncertainty range: 4–14%, global estimate 13% [*World Health Organization, 2012*]), previously treated cases at 13% of all incident cases (95% uncertainty range: 9–34%, global estimate 14% [*World Health Organization, 2012*]), and duration of TB disease at 1.2 years (95% uncertainty range 0.8–2.1, global estimate 1.4 years [*World Health Organization, 2012*]). Our model estimated that MDR-TB prevalence in previously treated cases was 15.4% (WHO estimate 20% [*World Health Organization, 2012*]), but unlike our model, WHO notifications often count failure and recurrence after default (in the same person) as two separate cases. At steady-state in the model, 80% of incident TB was due to recent infection rather than reactivation.

### Comparison of diagnostic strategies in the high incidence scenario

*Figure 3* shows the projected incremental 5-year cost and impact of each of nine selected diagnostic strategies (described in greater detail in the 'Materials and methods' section), in the high-incidence scenario. In general, both the cost and impact of targeted strategies (culture for retreatment, Xpert for

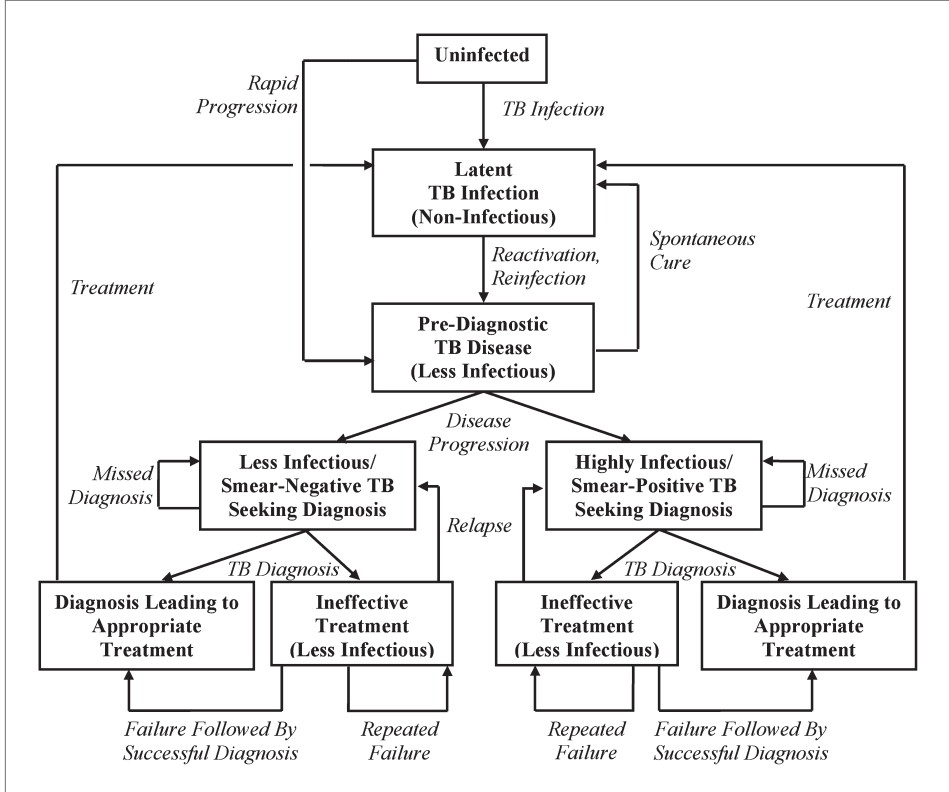

**Figure 2**. Transmission model of TB diagnosis. Boxes represent sub-populations in the model, and arrows represent rates of movement between those sub-populations. Parallel structures exist for: (a) HIV-infected vs HIV-uninfected; (b) never-treated vs previously treated (for TB); and (c) among TB-infected individuals, drug-susceptible vs isoniazid-monoresistant vs rifampin-resistant (including MDR). 'Pre-diagnostic' TB refers to individuals who are infectious but have not yet begun to seek care. Mortality occurs from all sub-populations (not shown), and at a higher rate among those with HIV and active TB.

HIV-positive, Xpert for smear-positive) on incidence were small relative to broader diagnostic strategies (Xpert for all, MODS/TLA, same-day Xpert). The incremental cost-effectiveness, in terms of cost per case averted (i.e., slope of the line from origin to each point in *Figure 3*), was similar across all strategies with the exception of same-day smear, which was cost-saving and had greater effectiveness than the baseline. Same-day microscopy remained cost-saving by year 5 as long as same-day results could be provided at less than five times the per-smear cost of routine results (i.e., <$10/test). Among the targeted strategies, Xpert for smear-positives was the most expensive but had the greatest impact on MDR-TB cases averted, whereas Xpert for HIV-infected individuals and culture for retreatment cases offered smaller gains at lower cost. Among the broad strategies, culture confirmation of rifampin-resistant tests on Xpert saved costs with little reduction in effectiveness. There was little difference between culture-confirmed Xpert and MODS/TLA, and same-day Xpert was the most expensive and most effective strategy.

## Comparison of diagnostic strategies across settings

The projected impacts and costs of the nine diagnostic strategies relative to the baseline strategy, in each of four selected epidemiological settings, are shown in *Figure 4*. In all four settings, the ranking of diagnostic strategies remained similar for all outcomes, although the cost of broader diagnostic strategies relative to targeted strategies fell substantially over 5 years in higher-incidence settings as the broader strategies generated declines in TB incidence. In the low-incidence setting, where a higher proportion of TB treatments are false-positive and more incident TB is also due to reactivation (60% of all new cases), the relative cost of improved TB diagnosis was the highest, while the relative impact was the least. In the high HIV setting, the impact of diagnostic interventions on TB incidence was

**Table 1.** Model input parameters*

| Parameter | Value | Reference(s)/Rationale |
|---|---|---|
| *TB and HIV Transmission* | | |
| Transmission rate, per smear-positive/highly infectious person-year | | Calibrated to user-defined TB incidence† |
| Proportional reduction in per-case transmission rate, MDR-TB | | Calibrated to user-defined MDR-TB prevalence† |
| Proportional reduction in fitness, isoniazid-monoresistant TB | 25% of MDR-TB reduction | Assumption |
| HIV incidence rate, per year | | Calibrated to user-defined HIV prevalence† |
| Relative transmission rate from smear-negative/less infectious TB | 0.22 | (*Behr et al., 1999*) |
| Proportion of pulmonary TB that is smear-positive/highly infectious | | |
| HIV-negative | 0.63 | (*Steingart et al., 2006a*; *Steingart et al., 2006b*) |
| HIV-infected | 0.50 | (*Getahun et al., 2007*) |
| *TB Progression* | | |
| Endogenous reactivation rate | | |
| HIV-negative | 0.0005/year | (*Horsburgh et al., 2010*) |
| HIV-infected | 0.05/year | (*Antonucci et al., 1995*) |
| Proportion of recent infections resulting in rapid progression | | |
| HIV-negative | 0.14 | (*Vynnycky and Fine, 1997*; *Dye et al., 1998*) |
| HIV-infected | 0.47 | 0.75 without ART, (*Daley et al., 1992*) |
| | | 75% reduction if on ART, (*Williams et al., 2010*) 50% ART coverage |
| Reduction in TB rapid progression probability due to latent TB infection (HIV-negative only) | 0.79 | (*Andrews et al., 2012*) |
| *TB Mortality and Resolution* | | |
| Life expectancy at age 15 | 55 years | (*World Bank, 2012*) |
| Annual mortality from HIV | 0.05/year | (*UNAIDS, 2012*) |
| Annual mortality from TB | | |
| HIV-negative, smear-positive/highly infectious | 0.23/year | (*Tiemersma et al., 2011*) |
| HIV-negative, smear-negative/less infectious | 0.07/year | (*Tiemersma et al., 2011*) |
| HIV-infected | 1.0/year | (*Corbett et al., 2003*; *Corbett et al., 2007*; *Wood et al., 2007*) |
| Rate of spontaneous TB resolution (HIV-negative only) | | |
| Smear-positive/highly infectious | 0.1/year | (*Tiemersma et al., 2011*) |
| Smear-negative/less infectious | 0.27/year | (*Tiemersma et al., 2011*) |
| *TB Treatment Outcomes and Emergence of Drug Resistance* | | |
| Probability of failure or relapse (within 1 year) | | |
| Drug-susceptible | 0.04 | (*World Health Organization, 2012*) |
| INH-monoresistant, first-line therapy | 0.21 | (*Menzies et al., 2009b*) |
| INH-monoresistant, retreatment or 2nd-line | 0.16 | (*Menzies et al., 2009b*) |
| MDR-TB, first-line or retreatment | 0.50 | (*Espinal et al., 2000*) |
| MDR-TB, second-line therapy | 0.30 | (*World Health Organization, 2010*) |

*Table 1. Continued on next page*

*Table 1. Continued*

| Parameter | Value | Reference(s)/Rationale |
|---|---|---|
| Proportion of one-year recurrence due to failure | | |
| Drug-susceptible | 0.14 | (*Lew et al., 2008*) |
| INH-monoresistant | 0.33 | |
| MDR-TB | 0.56 | |
| Probability of acquired drug resistance (per treatment course) | | |
| Susceptible becoming INH-monoresistant | 0.001 | (*Menzies et al., 2009a*; *Menzies et al., 2009b*) |
| Susceptible becoming MDR-TB | 0.002 | |
| INH-monoresistant becoming MDR-TB | 0.045 | |
| If treated with 2 effective drugs for >6 mos | 0.017 | |
| *Behavioral Parameters* | | |
| Infectious months before starting to seek care | | |
| HIV-negative | 9 months | (*Dowdy et al., 2013*) |
| HIV-infected | 1 month | (*Corbett et al., 2004*) |
| Diagnostic frequency while seeking care | 5.0/year | (*Storla et al., 2008*; *Sreeramareddy et al., 2009*) |
| Probability of treatment in a TB patient whose microbiological test is negative | 0.25 | (*Wilkinson et al., 2000*; *Dowdy et al., 2008*) |
| Loss to follow-up between diagnostic presentation and treatment initiation | | |
| Sputum smear or GXP (not same-day) | 0.15 | (*MacPherson et al., 2014*) |
| Culture (microcolony or commercial liquid) | 0.25 | (*Dowdy et al., 2008*) |
| Same-day diagnosis | 0 | Assumption |
| *Diagnostic Accuracy* | | |
| Sensitivity for smear-negative/less-infectious TB | | |
| Sputum smear microscopy | 0 | |
| Xpert MTB/RIF | 0.72 | (*Brownell et al., 2012*) |
| Culture (microcolony or commercial liquid) | 0.85 | (*Cruciani et al., 2004*; *Leung et al., 2012*) |
| Specificity for TB | | (*Steingart et al., 2006*; *Boehme et al., 2011*; *Leung et al., 2012*) |
| Sputum smear microscopy | 0.98 | |
| Xpert MTB/RIF | 0.98 | |
| Microcolony culture | 0.98 | |
| Sensitivity for drug resistance (if TB detected) | | |
| Microcolony culture (rifampin and isoniazid) | 0.98 | (*Minion et al., 2010*) |
| Xpert MTB/RIF (rifampin only) | 0.94 | (*Boehme et al., 2011*) |
| Specificity for drug resistance (if TB detected) | | |
| Microcolony culture (isoniazid) | 0.96 | (*Minion et al., 2010*) |
| Microcolony culture (rifampin) | 0.99 | (*Minion et al., 2010*) |
| Xpert MTB/RIF (rifampin) | 0.98 | (*Boehme et al., 2011*) |
| *Diagnostic Delay and non-TB Care-Seeking* | | |
| Days from presentation to treatment initiation | | |
| Sputum smear or Xpert MTB/RIF | 7 days | Assume 1 week |
| Microcolony or commercial liquid culture | 30 days | (*Boehme et al., 2011*) |

*Table 1. Continued on next page*

*Table 1. Continued*

| Parameter | Value | Reference(s)/Rationale |
|---|---|---|
| Months of therapy before a failing regimen will be changed, or before default and recurrence | 6 months | Assumption |
| Annual rate of diagnostic evaluation for TB, among people who do not have active TB | 0.01/year | 10% of suspects have TB, high-incidence setting |
| *Cost Parameters (user-defined; values below for comparison purposes only)* | | |
| Per-patient cost of TB therapy | | |
| First-line | US$500 | User-defined |
| Retreatment | US$1000 | (*Vassall et al., 2011*) |
| Second-line/MDR | US$5000 | (*Vassall et al., 2011*) |
| Outpatient visit (diagnosis or follow-up) | US$10 | (*Vassall et al., 2011*) |
| Per-test cost: | | |
| Sputum smear | US$2 | (*Vassall et al., 2011*) |
| Same-day sputum smear | US$10 | Assumption |
| Xpert MTB/RIF | US$15 | (*Vassall et al., 2011*) |
| Same-day Xpert MTB/RIF | US$30 | Assumption |
| Microcolony culture (with DST) | US$5 | (*Solari et al., 2011*) |
| Commercial liquid-media culture | US$20 | (*Vassall et al., 2011*) |
| Commercial liquid-media culture + DST | US$40 | (*Vassall et al., 2011*) |

*In the actual model program (**Supplementary file 1**), users can change any parameter based on local values.
†For reference, the transmission rate (in infections per person-year during diagnosis-seeking active TB) is 36.9 in the reference scenario, 14.0 in the low-incidence scenario, 25.4 in the high MDR scenario, and 12.9 in the high HIV scenario. Corresponding proportional reductions in MDR-TB transmission rate are 0.23, 0.23, 0.21, and 0.19; and HIV incidence estimates (per 1000 adult person-years) are 0.7, 0.6, 0.6, and 18.9.

diminished relative to the high-incidence setting, though the impact on TB mortality was similar. Additionally in this setting, the Xpert for HIV-positive strategy was substantially more costly, but also more effective, than in other settings.

## Sensitivity analyses

The impact of the 'Xpert for all' strategy on TB incidence (selected a priori as the primary outcome for sensitivity analysis) was most sensitive to three parameters, both in terms of absolute effects on impact estimates and partial rank correlation coefficients. These three parameters were: (1) the proportion of TB patients who would be empirically treated even if microbiologic testing yielded a negative result ('empiric treatment proportion'), (2) duration of infectiousness before seeking care ('pre-diagnostic delay'), and (3) rate of reactivation from latent infection to active disease ('reactivation rate'). For this latter parameter, in-depth investigation revealed that the key determinant was not the reactivation rate per se, but rather the proportion of active TB representing recent vs remote infection. If the empiric treatment proportion was increased from 25% to 37.5%, the projected reduction in TB incidence fell from 20% to 13%. By contrast, when only 12.5% of false-negative patients were started on therapy, 'Xpert for all' achieved a 31% reduction in incidence. Corresponding reductions in incidence with 'Xpert for all' (20% at baseline) included: 27% if pre-diagnostic delay was shortened from 9 months to 4.5 months, 14% if pre-diagnostic delay was lengthened to 13.5 months, 15% if the reactivation rate was doubled from 0.05% to 0.1% per year, and 23% if reactivation was halved to 0.025% per year. The projected 5-year reduction in incidence for this strategy did not fall outside the range of 12–25% under variation of any other model parameter, up to 50% of that parameter's baseline value (*Table 1*).

Both costs and incremental costs were more sensitive to the cost of TB treatment than the cost of diagnostics. For example, doubling the cost of first-line therapy (from $500 to $1000 per person) augmented the incremental year 1 costs for 'Xpert for all' from a 57% increase over baseline to a 77% increase, and doubling the cost of MDR therapy (from $5000 to $10,000 per person) generated an

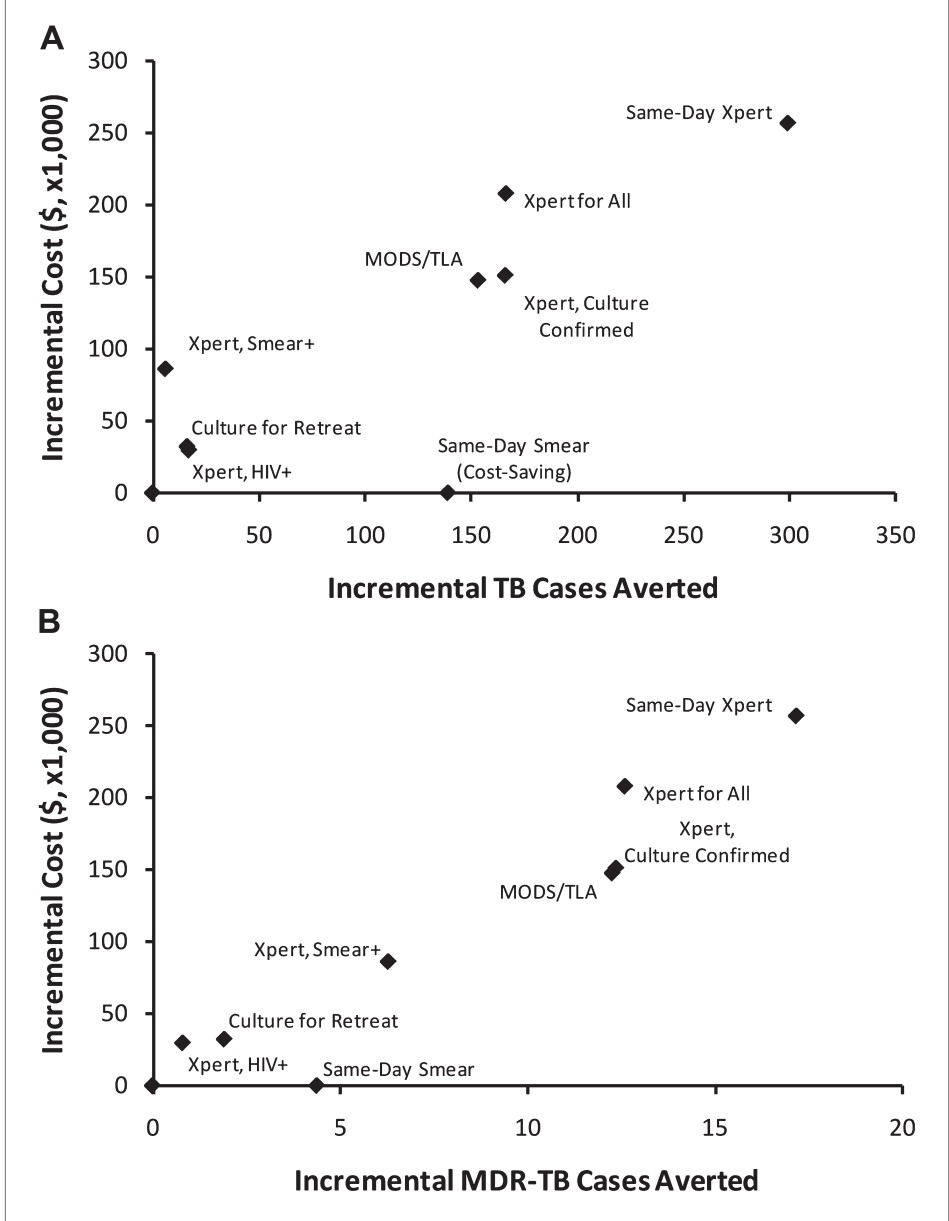

**Figure 3**. Incremental 5-year cost and impact of TB diagnostic strategies, high-incidence setting. Shown are cumulative projected 5-year costs and impact (averted TB cases [panel **A**] or MDR-TB cases [panel **B**]) of each diagnostic strategy described in the Introduction, incremental to the baseline strategy, per 100,000 population. Strategies with greater impact appear to the right on the x-axis; more costly strategies appear higher on the y-axis. The same-day smear strategy is cost-saving but shown at an incremental cost of $0 for simplicity.

86% increase in incremental year 1 costs, whereas doubling the unit cost of Xpert (from $15 to $30) only resulted in a 72% increase. Unit costs other than those for first-line treatment, MDR treatment, and the diagnostic modality under study in each scenario were not important determinants of incremental costs.

## Discussion

To date, most transmission models of infectious disease control interventions present results that are not directly usable by decision makers because they are not customizable to local conditions. We present a flexible, user-friendly model of TB diagnosis and transmission that allows users without

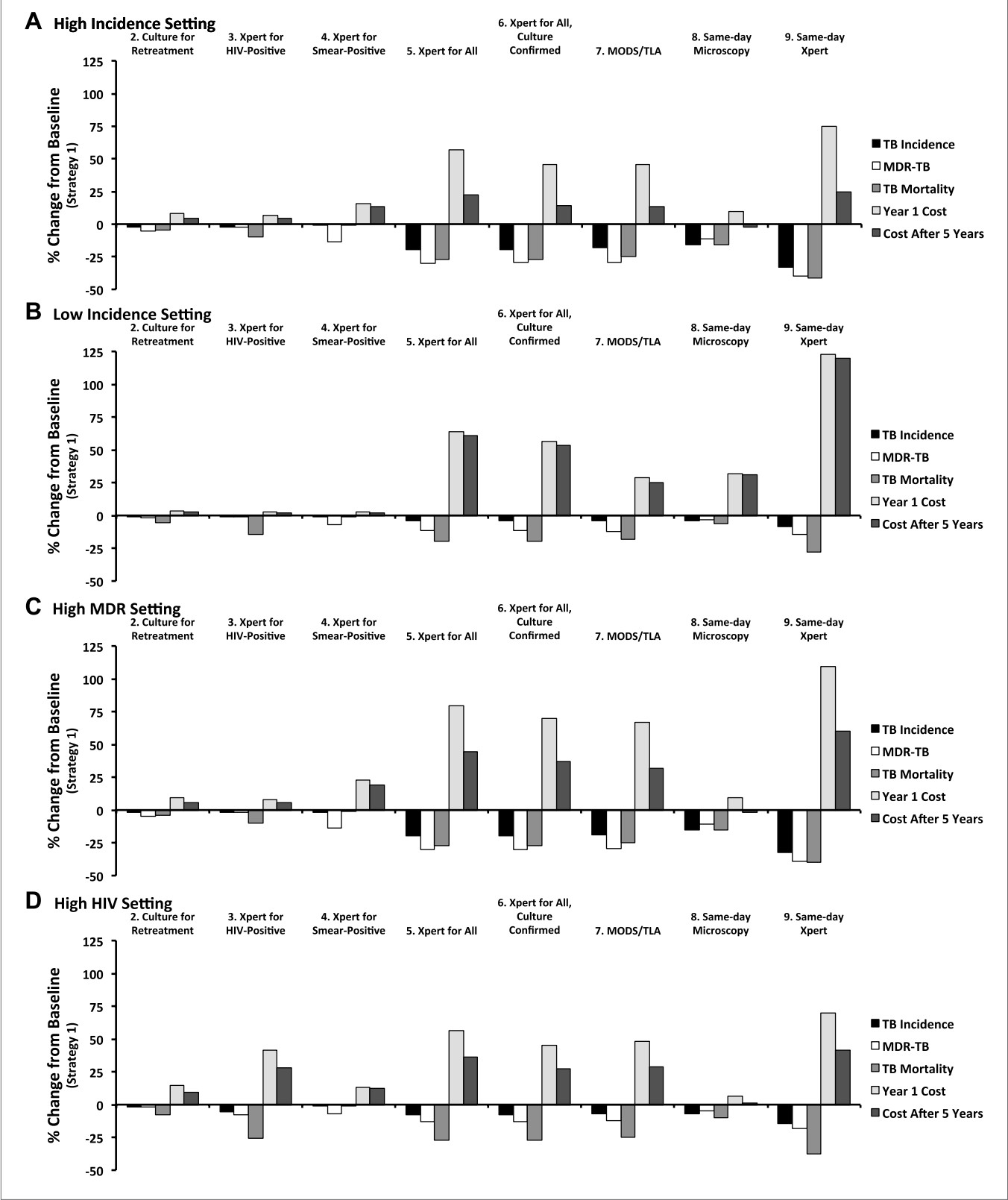

**Figure 4**. Relative impact of diagnostic strategies in emblematic settings. Shown are projected changes in TB incidence, MDR-TB incidence, TB mortality, and costs (in Year 1 and Year 5 after immediate implementation), relative to baseline (Strategy 1) after implementing each of the diagnostic

*Figure 4. Continued on next page*

*Figure 4. Continued*

strategies described in the text. Epidemiological outcomes are measured at the end of Year 5. Panel **A** (high incidence) shows a setting with
TB incidence of 250 per 100,000/year, stable MDR-TB prevalence of 3.7% among new cases, adult HIV prevalence of 0.83%, and cost of $500 to treat one
case of TB with first-line therapy. In panel **B** (low incidence), the TB incidence is reduced to 8.3 per 100,000/year (implemented by gradual decline in
incidence over 50 years). In panel **C** (high MDR), MDR-TB prevalence among new cases is set at 3.7% in the beginning of year 1, increasing to 10.7% by
the end of year 5. In panel **D** (high HIV), adult HIV prevalence is set to 20% and TB incidence is set to 500 per 100,000/year.

modeling expertise to define an epidemiologic setting (according to TB incidence, MDR-TB prevalence, and HIV prevalence) and unit costs, evaluating various diagnostic strategies in that setting in terms of population-level costs and impact. While this model cannot precisely replicate the epidemiological situation in any given location, it applies a standardized methodology across a wide range of settings, thereby illustrating important interactions between epidemiological parameters and projected impact. We provide both a web interface for rapid calculations and full model code whereby users can change any model parameter for 'personalized' sensitivity analysis. Our model results suggest that the rank-ordering of diagnostic strategies may be relatively stable across epidemiological settings, but that the actual population-level costs and impact differ dramatically. This model also serves as an example of how epidemiologists can provide decision-makers across a wide variety of local settings with rapid access to customizable 'first-pass' projections of cost and impact from transmission models without the need to construct tightly fitted models to represent all epidemiological settings.

Our results provide important guidance to TB decision-makers. Specifically, in settings where little additional up-front investment is possible (<25% increase in TB control budget), same-day microscopy has the greatest impact on TB incidence with no increase in overall cost at 5 years, provided that same-day microscopy can be feasibly delivered for less than $10 per patient. For settings in which containing MDR-TB is the most important consideration, Xpert for smear-positives has the greatest effectiveness for this outcome, but at substantial price and very little impact on overall TB incidence and mortality. Xpert for HIV-positives and culture for retreatment cases both offer meaningful, albeit small, gains; the cost and impact of these strategies are the highest in HIV-endemic settings. In settings where more initial investment is possible (about 50% increase in TB control budget), broader scale-up of either Xpert or MODS/TLA for all TB diagnosis can offer substantially greater benefits, both in terms of reduced incidence (often 10 times more TB cases averted than with the more narrowly-targeted strategies above) and long-term costs (in that the incremental cost of these strategies declines greatly by Year 5). In general, culture confirmation of positive Xpert results before committing to a course of second-line therapy is preferred. Finally, where the greatest impact is sought, combination of Xpert for all plus infrastructure for same-day diagnosis achieves this aim in all settings, but at the highest cost.

Although this model represents a highly simplified framework, it compares well to other global estimates to which it was not fit (e.g., WHO estimates of TB mortality, previously treated TB, HIV-associated TB, and MDR-TB prevalence among previously treated cases). Its results are also similar to those of other mathematical models of TB diagnosis that are fit to specific locations. For example, *Menzies et al. (2012)* modeled scale-up of Xpert for people living with HIV in southern Africa, estimating a 6% reduction in incidence, 21% reduction in TB mortality, 25% reduction in MDR-TB incidence that declines to <10% over a longer time frame, and 40% increase in costs (not including costs of antiretroviral therapy). Our corresponding estimates for the Xpert for HIV-positive strategy in the high-HIV setting are 5% reduction in incidence, 25% reduction in TB mortality, 7% reduction in MDR-TB incidence, and increase in costs from 41% (year 1) to 28% (year 5). For purposes of deciding between alternative strategies of scaling up TB diagnostics, the estimates from these two models are likely to provide similar guidance.

For any such simplified transmission model to have an impact on decision-making, it is important to consider who the eventual users of such a model might be, and to develop the model in consultation with those groups. In this case, we identified a number of potential end-user organizations including technical assistance agencies (e.g., KNCV Tuberculosis Foundation), funding bodies (e.g., Global Fund for AIDS, Tuberculosis, and Malaria), non-governmental organizations (e.g., Medecins Sans Frontiers), National Tuberculosis Programs, and regional offices of the World Health Organization. We then invited representatives of these organizations to attend a 1-day workshop (April 2014, The Hague, The

Netherlands) describing methods and challenges related to modeling of TB diagnostics in general, including this transmission model in particular. We developed 'hands-on' exercises for participants to better learn the model and solicited specific feedback, which is being incorporated into the model structure and web interface in ongoing fashion. We next intend to develop a series of informal 'case studies' whereby the use of this model for in-country decision-making can be demonstrated and disseminated.

As with any modeling analysis, this research has important limitations. In order to provide sufficient flexibility and generalizability, we make a number of strong assumptions that include a constant population, homogeneous mixing, no change in parameter values over time, and simplistic incorporation of HIV and drug resistance. Without making such simplifying assumptions, it is impossible to deliver a flexible modeling framework that can generate transparent, customizable, rapid results (i.e., without complex statistical fitting to each individual epidemiological scenario). This model can replicate user-defined TB incidence, MDR-TB prevalence, and HIV incidence but does not describe the breakdown of these values in key subpopulations (e.g., congregate settings or geographical 'hotspots'), nor does it incorporate operational aspects of the TB diagnostic system in any one setting. Thus, this model does not ameliorate the need for more detailed models in settings where precise estimates are needed; rather it provides access to 'first-pass' estimates in settings (i.e., the vast majority of local decision-makers) where such tightly calibrated model projections are not available. This model focuses on transmission dynamics and thus does not include pediatric TB and extrapulmonary TB; these largely non-infectious disease manifestations remain very important components of the TB epidemic that are not captured here. We validate our model against global estimates and other models; ideally, further validation would be performed over time using field data across a wide variety of settings. Our model allows users to define three key epidemiological parameters (as well as other model assumptions within the program), but data to inform even these three parameters—as well as unit costs—are unlikely to be available on sub-national levels. As a result, users wishing to adapt the model to a smaller geographic scale will need to perform additional data-gathering exercises to inform these estimates if they wish to maximize the utility of the model.

Nevertheless, even if high-quality data are not available at the local level, this model allows decision-makers to estimate epidemiological and economic values according to reasonable assumptions (e.g., comparison of TB notification rates or budget line-items to those in the published literature or at the national level), vary those assumptions in real-time, and obtain corresponding projections of comparative impact that incorporate the best available current data on epidemiological or natural history parameters (e.g., TB progression and reactivation). Future efforts might also provide flexibility to specify operational characteristics (e.g., health system capacity) as well.

A final concern is that, by providing users the ability to specify TB incidence, MDR-TB prevalence, and adult HIV prevalence, a number of scenarios can be created (e.g., low TB incidence and very high adult HIV prevalence) that are not epidemiologically realistic. Although there is danger in allowing uninformed users to make projections for such scenarios (and the model will reject or alert users to highly implausible values), we believe that this risk is outweighed by the benefit of providing full flexibility to model epidemiological scenarios (e.g., sub-district level data) that will never be captured by a limited number of closely-calibrated TB transmission models.

In summary, we have created a flexible modeling framework that allows users without modeling expertise to generate simulated populations with locally relevant values for TB incidence, MDR-TB prevalence, adult HIV prevalence, and TB treatment costs. By comparing an array of diagnostic options across emblematic epidemiological scenarios, we provide guidance to decision-makers who seek to ascertain the optimal diagnostic strategy to achieve their selected disease control targets, and to do so using a standardized methodology. Success in the fight against infectious disease generally, and TB specifically, depends on our ability to place global knowledge in the hands of local decision-makers, enabling them to choose those interventions that are likely to have the greatest impact, given existing resources and local epidemiological realities. This flexible modeling framework of diagnostic interventions takes an important step in that direction.

## Materials and methods

### Transmission model

Using previously published models of TB diagnostics as a guide (*Dye et al., 1998*; *Abu-Raddad et al., 2009*; *Menzies et al., 2012*; *Dowdy et al., 2013*), we constructed a transmission model of TB using

ordinary differential equations. This model categorizes patients according to HIV status (positive or negative), TB treatment status (never treated or previously treated), TB disease status (as shown in *Figure 2*), and among those who are infected with TB, drug resistance status (susceptible, isoniazid-monoresistant, and rifampin-resistant including MDR), and level of infectiousness (smear-negative/less-infectious and smear-positive/highly infectious). Individuals enter the model at age 15, and TB with no pulmonary component (i.e., not infectious) is not included. For purposes of transparent communication, we chose a population size of 100,000 (to match standard reporting of TB outcomes) and assumed a constant population with no net population growth or immigration/emigration. After constructing the transmission framework, we used decision analysis to estimate (a) the probability of each diagnostic outcome; (b) the diagnostic delay; and (c) the cost of TB diagnosis and treatment under each of the nine diagnostic strategies, assuming immediate implementation at the beginning of a given year ('Year 1'). Two separate authors (DWD and PJD) independently coded the model; these models gave comparable results.

## Model initiation

After setting probabilities and costs under each diagnostic strategy, we then created a flexible modeling structure capable of generating epidemiological scenarios as a function of three variables, which the user specifies: HIV prevalence (assuming a global mean level of antiretroviral therapy coverage), TB incidence, and MDR-TB prevalence among new TB cases. We accomplished this by allowing three key parameters to vary across model scenarios: annual HIV incidence, rate of TB transmission per smear-positive/highly infectious person-year, and relative per-case transmission rate of rifampin-resistant TB. We also allow users to specify all relevant unit costs for TB diagnosis and treatment; other model parameters were estimated from existing literature (*Table 1*). We then created a program that numerically generates a unique steady-state population meeting the user-defined values; this population serves as the baseline strategy (Strategy 1 above) at the beginning of the time frame under evaluation. The program has flexibility to create its steady-state population 50 years in the past, allowing it to replicate the protracted, slow declines in TB incidence as seen in many lower-incidence settings; this is done automatically for any scenario with a target TB incidence less than 50 per 100,000/year.

## Model compartments

The mathematical model consists of the following TB compartments:

- $U_{hp}$, Uninfected
- $L_{hdp}$, Latently infected
- $E_{hdp}$, Early active (infectious status $i = 0$)
- $A_{hdip}$, Late active
- $P_{hdip}$, Active 'pre-treatment': diagnosis in progress, will lead to appropriate therapy
- $I_{hdip}$, Active 'inappropriate treatment': receiving therapy that ends in default or failure

In these compartments, the subscript $h$ refers to HIV status ($h = 0$ if HIV-uninfected, 1 if HIV-infected), $d$ refers to drug resistance status ($d = 0$ if drug-susceptible, 1 if isoniazid [INH]-monoresistant, and 2 if multidrug-resistant [MDR]), $i$ refers to infectious status ($i = 0$ if smear-negative/less infectious and 1 if smear-positive/highly infectious), and $p$ refers to previous treatment status ($p=0$ if never treated, 1 if previously treated). Infectious status can be conceptualized as an individual's sputum smear status, if two smears were to be performed in a quality-assured laboratory at any given point in time.

## Model structure

The model assumes an adult population of stable size with no immigration or emigration: the number of individuals entering the uninfected compartment $U$ is set as equal to the number who die (whether from TB or other causes) from all other compartments. Pediatric and purely extrapulmonary TB are not explicitly considered because the diagnostic considerations for these manifestations are different. In the short-term, however, to the extent that these forms of TB are non-infectious and equally fatal as adult pulmonary TB, their effects on TB incidence and mortality may be approximated by dividing the model's projected incidence and mortality by (1 − proportion of TB that is not adult pulmonary), to obtain a new incidence/mortality estimate. Thus, if 20% of all TB in a given location is extrapulmonary or paediatric, the rough projected total TB incidence would be (projected TB incidence)/(0.8).

In this model, we consider latent TB infection to be asymptomatic and non-infectious, with a constant rate of reactivation and ongoing risk of exogenous reinfection leading to active TB; individuals successfully treated for TB are assumed to return to this compartment upon initiation of effective therapy (i.e., therapy that will result in completion, with no relapse for 2 years). Upon developing active TB, individuals enter a 'pre-diagnostic' phase that is characterized by a low level of infectiousness and mortality (equivalent to smear-negative TB) and during which individuals do not actively seek diagnosis. The duration of this phase (9 months) was selected a priori based on an existing model (*Dowdy et al., 2013*) in which the total duration of disease after incorporating this phase reflected the global ratio of prevalence to incidence, as estimated by the World Health Organization. We compared the total duration of disease to this ratio as part of our model validation and assumed that this 'pre-diagnostic' phase is much shorter for HIV-infected vs HIV-uninfected individuals. Upon completing this 'pre-diagnostic' phase, individuals progress to a diagnosis-seeking phase of active disease, which is characterized by separation into highly infectious ('smear-positive') and less infectious ('smear-negative') compartments. Among HIV-uninfected individuals, the highly infectious compartment also carries higher mortality risk. Diagnosis-seeking active TB implies active seeking of diagnosis at a defined rate; the probability that any single diagnostic attempt will result in effective therapy is calculated as a function of diagnostic sensitivity, probability of empiric therapy (i.e., without bacteriological confirmation), prior treatment status, and losses to follow-up, as described below. Each diagnostic attempt, if successful, leads either to effective therapy (which is initiated after a defined diagnostic delay) or to ineffective therapy (defined as leading to failure or default). In order to focus on differences between the nine selected strategies above in a tractable framework, we subsumed all other diagnostic tests and procedures (e.g., chest X-ray, antibiotic trials) as a probability of non-microbiologic diagnosis, without attempting to specify the associated cost or diagnostic delay. Once effective therapy is initiated, it is assumed to immediately render the individual non-infectious, with no residual risk of mortality. Upon initiation of ineffective therapy, individuals are assumed to remain infectious (at the 'smear-negative'/less infectious level) for a defined period before either failing (followed by another round of therapy, which can be either appropriate/curative or ineffective) or default. Reasoning that default will occur, on average, at the midpoint between receipt of fully-ineffective and fully-effective therapy, half of defaulters are presumed to develop recurrent TB (which is assumed to occur immediately), while the other half return to the latent TB compartment (from which reactivation or reinfection remains possible). All individuals who relapse within 1 year are included as failures; thus, no specific parameter for relapse is incorporated. Individuals who are effectively treated, or whose disease is contained without therapy, return to the latent compartment following the convention of other TB models (*Dye et al., 1998*; *Abu-Raddad et al., 2009*).

## Role of HIV coinfection

As the goal of this model is to focus on TB-related interventions, HIV is modeled as occurring at a defined annual incidence, calibrated to achieve a given user-defined prevalence at baseline. We do not explicitly model HIV infection in dynamic fashion (i.e., the HIV incidence rate does not depend on the number of HIV-infected individuals in the model). HIV infection is assumed to affect all parameters related to TB disease, including the level of immune protection afforded by latent infection (assumed zero if HIV-infected), mortality rate (increased), rate of reactivation from latent TB (increased), duration of 'pre-diagnostic' TB (decreased), and risk of 'primary' progression to active disease upon infection (increased). For purposes of maintaining a simple model structure, we do not explicitly model CD4 counts or antiretroviral therapy (ART), but instead assume the global average of ART coverage, as estimated by *UNAIDS (2012)*. We weight all HIV-related parameters according to this estimated probability of ART receipt; this probability can be modified by users.

## Drug resistance

We assume infection with TB strains of three different drug resistance levels: fully susceptible, INH-monoresistant, and MDR. Dual infection with multiple strains is not considered in this model, but superinfection (i.e., reinfection of a latently infected individual with a different strain, resulting in primary progression to disease with the reinfecting strain) is allowed, as is acquisition of resistance (i.e., change of strain from more susceptible to less susceptible) as a result of therapy.

### Individuals without TB

A key consideration with any TB diagnostic strategy is the role of the diagnostic test as applied to individuals who do not have TB (i.e., specificity). We included this element by considering that a small proportion of the population without TB would present with TB-like symptoms each year. This proportion (selected such that 10% of individuals being evaluated in the high-incidence setting actually have underlying TB) remains constant across all scenarios, such that the pre-test probability of TB is higher in settings of high TB incidence, and declines over time as successful TB control strategies are employed. Individuals without TB who are (inappropriately) treated for TB incur costs of TB therapy and are also marked as 'previously treated' for purposes of diagnostic evaluation in the future.

### Economic evaluation

Economic parameters are estimated using a unit-costing approach, whereby the unit cost of a TB diagnostic attempt and a TB treatment course (separately for first-line, category two, and second-line therapy) is enumerated under each scenario, and this cost is multiplied by the number of diagnostic attempts and treatment courses performed. For simplicity, we adopt the perspective of a TB control program for our costing; additional costs of HIV care and general health services (e.g., hospitalization) are not included. The decision tree below is used to estimate the cost per diagnostic attempt or treatment course, conditional on an individual's HIV, drug resistance, and previous treatment status. Individuals who default or die on therapy are assumed to incur half the cost of a treatment course. Given the short time horizon (5 years), the focus on costs and outcomes (rather than cost-effectiveness per se), and the desire to compare costs in year 1 and at the end of year 5 in equivalent terms, we did not discount future costs or outcomes for this analysis. All costs are reported in US dollars, assuming the year of costs that is specified by the user.

### Decision analysis: probability of diagnostic success

Under each scenario, we use decision analysis to ascertain the following quantities related to each diagnostic attempt:

- Probability of receiving successful treatment.
- Probability of receiving ineffective treatment (resulting in failure or default, including the probability of acquired resistance).
- Cost of treatment (conditional on whether treatment is successful or ineffective).
- Cost of diagnosis.
- Diagnostic delay incurred before treatment initiated.

These quantities are calculated conditional on each patient's infectious (smear) status, drug resistance status, HIV status, and prior treatment status. These probabilities are calculated for each diagnostic attempt, with the result fed back into the transmission model for purposes of appropriately allocating flows between compartments. Inputs into the decision model include the probabilities of failure/recurrence, probability of empiric therapy, loss to follow-up before treatment, diagnostic accuracy, diagnostic delay, and economic parameters as shown in *Table 1*. Outputs from the decision tree appear as parameters in the model, as described in *Table 2* and the following equations.

### Outcomes and sensitivity analysis

Our primary outcomes under each scenario were TB incidence, TB mortality, MDR-TB incidence, and incremental TB diagnostic and treatment costs (during year 1 and at the end of the 5-year period) relative to the baseline strategy. By giving flexibility to change model inputs, we provide users the ability to conduct any sensitivity analysis desired. However, for illustrative purposes, we also conducted a series of one-way sensitivity analyses in which each model parameter in *Table 1* was varied by ±50% of its listed value (for proportions, 50% of the difference between the value and either zero or one). Our primary outcome for sensitivity analysis was the change in TB incidence, comparing the 'Xpert for all' strategy to baseline in the high incidence setting.

We also conducted multivariable uncertainty analyses by calculating partial rank correlation coefficients (*Kendall, 1942*) between each natural history parameter and the outcomes of TB incidence, TB mortality, and 5-year costs. In addition, we constructed 95% uncertainty intervals around our estimates of outcomes in each individual country by simultaneously varying each model parameter by ±10% over a uniform distribution and each target value (i.e., TB incidence, HIV prevalence, and MDR-TB prevalence) over its reported

**Table 2.** Model parameters and symbolic representations

| Parameter | Representation | Baseline value (see *Table 1*) |
|---|---|---|
| Transmission rate (transmission events per highly infectious person-year) | $\beta$ | Calibrated to TB incidence |
| Proportional reduction in per-case transmission rate | | |
| Drug-susceptible TB | $\varphi_0$ | 1.0 |
| Isoniazid-monoresistant TB | $\varphi_1$ | 25%* of $\varphi_2$ |
| MDR-TB | $\varphi_2$ | Calibrated |
| HIV incidence rate, per year | $\theta$ | Calibrated to HIV prevalence |
| Relative transmission rate from smear-negative/less infectious TB | $\zeta$ | 0.22 |
| Proportion of pulmonary TB that is smear-positive/highly infectious | | |
| HIV-negative | $\psi_0$ | 0.63 |
| HIV-infected | $\psi_1$ | 0.50 |
| Endogenous reactivation rate, per year | | |
| HIV-negative | $\varepsilon_0$ | 0.005 |
| HIV-infected | $\varepsilon_1$ | 0.05 |
| Proportion of recent infections resulting in rapid progression | | |
| HIV-negative | $\pi_0$ | 0.14 |
| HIV-infected | $\pi_1$ | 0.47 |
| Reduction in TB rapid progression probability due to latent TB infection | | |
| HIV-negative | $\iota$ | 0.79 |
| HIV-infected | Not included | 0 |
| Baseline mortality rate, per year | $\mu_{bl}$ | 1/55 = 0.018 |
| Additional HIV-related mortality rate, per year | $\mu_h$ | 0.05 |
| Additional untreated TB-related mortality rate, per year | | |
| HIV-negative, smear-positive/highly infectious | $\mu_{t1}$ | 0.23 |
| HIV-negative, smear-negative/less infectious | $\mu_{t0}$ | 0.07 |
| HIV-infected | $\mu_{th}$ | 1.0 |
| Rate of spontaneous TB resolution, per year | | |
| Smear-positive/highly infectious | $v_1$ | 0.1 |
| Smear-negative/less infectious | $v_0$ | 0.27 |
| HIV-infected | Not included | 0 |
| Rate of starting diagnosis-seeking in active TB, per year | | |
| HIV-negative | $\delta_{e0}$ | 1.33 (9 months) |
| HIV-infected | $\delta_{e1}$ | 12 (1/month) |
| Rate of progression: ineffective therapy to repeat therapy (failure) or active TB (relapse), per year | $\delta_f$ | 6/12 = 0.5 |
| Rate of diagnostic evaluation for TB, per year | | |
| Late active TB | Input into decision tree | 5.0 |
| No active TB | $\tau_0$ | 0.01 |
| Decision tree outputs (in addition to unit costs): | | Vary by intervention |
| Successful diagnosis rate of late active TB, per year | $\sigma_{hdip}$ | |
| Rate of movement from successful diagnosis to treatment (1/diagnostic delay), per year | $\rho_{hdip}$ | |

*Table 2. Continued on next page*

*Table 2. Continued*

| Parameter | Representation | Baseline value (see *Table 1*) |
|---|---|---|
| Ineffective diagnosis rate of late active TB, per year | $\kappa_{hdip}$ | |
| Rate of diagnosis and treatment leading to new resistance, per year | | |
| susceptible to INH-monoresistant | $\alpha si_{hip}$ | |
| susceptible to MDR | $\alpha sm_{hip}$ | |
| INH-monoresistant to MDR | $\alpha im_{hip}$ | |

*Calculated such that $(1-\varphi_1) = 0.25*(1-\varphi_2)$.

uncertainty range. In this fashion, we constructed 10,000 simulations using Latin Hypercube Sampling (*McKay et al., 1979*) and took 95% uncertainty ranges as the 2.5[th] and 97.5[th] percentiles of outcomes in these simulations; these ranges are provided in the web-based version of the model for each country.

## Model equations

Rates of flow between compartments are governed by the system of ordinary differential equations listed in *Equations 1–6*.

We first define the forces of infection (according to resistance status) and total mortality for simplicity.

## Forces of infection ($\lambda_d$)

$$\lambda_d(t) = \left[\frac{\beta}{N(t)}\right] * \varphi_d *$$

$$\left\{\zeta * \sum_{h,p}\left[E_{hdp}(t) + A_{hd0p}(t) + P_{hd0p}(t) + I_{hd0p}(t)\right] + \sum_{h,p}\left[A_{hd1p}(t) + P_{hd1p}(t)\right]\right\}$$

Thus, TB infection is modelled as a density-dependent process, a function of the transmission rate ($\beta$), total number of individuals in the population $N(t)$, number of individuals with 'less infectious' TB ($E$, early active; $A_{hd0p}$, late active smear-negative; $P_{hd0p}$, 'pre-treatment' smear-negative; and $I$, active on ineffective treatment) weighted by the relative transmission rate $\zeta$, and the number of individuals with fully infectious TB ($A_{hd1p}$, late active smear-positive; $P_{hd1p}$, 'pre-treatment' smear-positive). Three separate forces of infection are calculated for the three separate strains of drug resistance, with relative transmission weights of $\varphi_d$. We use $\lambda_{tot}$ to denote the sum of these three forces.

## Total mortality ($\mu_{tot}$)

$$\mu_{tot}(t) = \mu_{bl} * N(t)$$

$$+ \mu_h * \sum_p\left\{U_{1p}(t) + \sum_d\left[L_{1dp}(t) + E_{1dp}(t) + \sum_i\left(A_{1dip}(t) + P_{1dip}(t) + I_{1dip}(t)\right)\right]\right\}$$

$$+ \mu_{t0} * \sum_{p,d}\left\{E_{0dp}(t) + A_{0d0p}(t) + P_{0d0p}(t) + \sum_i\left(I_{0dip}(t)\right)\right\}$$

$$+ \mu_{t1} * \sum_{p,h,d}\left\{A_{0d1p}(t) + P_{0d1p}(t)\right\}$$

$$+ \mu_{th} * \sum_{p,d}\left\{E_{1dp}(t) + \sum_i\left(A_{1dip}(t) + P_{1dip}(t) + I_{1dip}(t)\right)\right\}$$

Thus, total mortality is the sum of:

- baseline mortality $\mu_{bl}$ (experienced by all individuals $N$),
- HIV-associated mortality $\mu_h$ (experienced by all individuals with HIV, $h = 1$),

- 'less infectious' (i.e., lower) TB-associated mortality $\mu_{t0}$ (experienced by all HIV-uninfected individuals with early active TB, $E$, smear-negative active and 'pre-treatment' TB, $A_{0d0p}$ and $P_{0d0p}$, and ineffectively treated TB, $I_{0dip}$),
- 'highly infectious' (i.e., higher) TB-associated mortality $\mu_{t1}$ (experienced by all HIV-uninfected individuals with smear-positive active and 'pre-treatment' TB, $A_{0d0p}$ and $P_{0d0p}$), and
- TB/HIV–associated mortality $\mu_{th}$ (experienced by all HIV-infected individuals with any form of TB).

## Uninfected compartment (U)

$$\frac{dU_{hp}(t)}{dt} = I_{h=0,p=0} * \mu_{tot}(t) - \left[\lambda_0(t) + \lambda_1(t) + \lambda_2(t) + \mu_{bl}(t)\right] * U_{hp}(t)$$

$$-I_{h=0} * \left[\theta * U_{0p}(t)\right] + I_{h=1} * \left[\theta * U_{0p}(t) - \mu_h * U_{1p}(t)\right]$$

$$- I_{p=0} * \left[\tau_0 * (1-s_h) * U_{h0}(t)\right] + I_{p=1} * \left[\tau_0 * (1-s_h) * U_{h0}(t)\right] \qquad (1)$$

where $\mu_{tot}$ is the sum of all mortality (to maintain a constant population), $\lambda_d$ is the force of infection for a given drug resistance strain, $\mu_{bl}$ is the baseline mortality rate, $\mu_h$ is the HIV-specific mortality rate, $I_{eq}$ is an indicator function (= 1 if the condition $eq$ is met, 0 otherwise), $\theta$ is the HIV incidence rate, $\tau_0$ is the rate of seeking diagnosis among people without TB, and $s_h$ is the specificity of the diagnostic test. Thus, uninfected individuals exit through infection and death, acquire HIV according to the HIV incidence rate, and become previously treated for TB (inappropriately) according to the specificity of the test. The HIV-uninfected, not previously treated compartment is replenished at a rate that matches total mortality and thereby maintains a constant population.

## Latently infected compartment (L)

$$\frac{dL_{hdp}(t)}{dt} = \lambda_d(t) * (1-\pi_h) * \left\{U_{hp}(t) + \sum_d \left[L_{hdp}(t) * (1-I_{h=0} * \iota)\right]\right\}$$

$$+ I_{p=1} * \sum_{i,p}\left[\rho_{hdip} * P_{hdip}(t)\right] + I_{h=0} * \left\{v_0 * E_{0dp} + \sum_i \left[v_i * \left(A_{0dip} + P_{0dip} + I_{0dip}\right)\right]\right\}$$

$$- \left\{\left[\left(\lambda_0(t) + \lambda_1(t) + \lambda_2(t)\right) * (1-I_{h=0} * \iota) + \varepsilon_h + \mu_{bl} + I_{h=1} * \mu_h\right] * L_{hdp}(t)\right\}$$

$$- I_{h=0} * \left[\theta * L_{0dp}(t)\right] + I_{h=1} * \left[\theta * L_{0dp}(t)\right] \qquad (2)$$

where $\lambda_d$ is the strain-specific force of infection, $\pi_h$ is the proportion of recent infections that progress rapidly to active TB, $\iota$ is the relative reduction in rapid progression after infection among people with latent TB, $I_{eq}$ is an indicator function (= 1 if the condition $eq$ is met, 0 otherwise), $\rho_{hdip}$ is the rate of treatment after successful diagnosis is initiated, $v_i$ is the spontaneous recovery rate, $\varepsilon_h$ is the endogenous reactivation rate, $\mu_{bl}$ is the baseline mortality rate, $\mu_h$ is the HIV-specific mortality rate, and $\theta$ is the HIV incidence rate. Thus, individuals enter the latently infected compartment through initial TB infection (without rapid progression), reinfection (without rapid progression, and accounting for immune protection), initiation of successful treatment, or spontaneous resolution. Individuals completing treatment only enter the previously treated compartment (p=1). Individuals exit this compartment through TB reinfection, endogenous reactivation, and death, and they acquire HIV infection at a constant rate.

## Early active compartment (E)

$$\frac{dE_{hdp}(t)}{dt} = \lambda_d(t) * \pi_h * \left\{U_{hp}(t) + \sum_d \left[L_{hdp}(t) * (1-I_{h=0} * \iota)\right]\right\} + L_{hdp}(t) * \varepsilon_h$$

$$- \left\{\left[\mu_{bl} + I_{h=0} * (\mu_{t0} + v_0 + \delta_{e0}) + I_{h=1} * (\mu_h + \mu_{th} + \delta_{e1})\right] * E_{hdp}(t)\right\}$$

$$- I_{h=0} * \left[\theta * E_{0dp}(t)\right] + I_{h=1} * \left[\theta * E_{0dp}(t)\right] \qquad (3)$$

where $\lambda_d$ is the force of infection, $\pi_h$ is the proportion of recent infections that progress rapidly to active TB, $\iota$ is the relative reduction in rapid progression after infection among people with latent TB, $I_{h=0}$ is an indicator function of HIV status (= 1 if $h = 0$, 0 otherwise), $\varepsilon_h$ is the endogenous reactivation rate, $\mu_{bl}$ is the baseline mortality rate, $\mu_{t0}$ is the TB-specific mortality rate for less-infectious TB, $v_0$ is the spontaneous recovery rate for less-infectious TB, $\delta_{eh}$ is the HIV-specific rate of progression to late active TB, $\mu_h$ is the HIV-specific mortality rate, $\mu_{th}$ is the TB-specific mortality rate for people living with HIV, and $\theta$ is the HIV incidence rate. Thus, individuals enter this compartment through rapid progression of recent infection or endogenous reactivation of latent infection. They exit through progression to late active disease, spontaneous cure (if HIV-uninfected), or death, and they acquire HIV infection at a constant rate.

## Late active compartment (*A*)

$$\frac{dA_{hdip}(t)}{dt} = \delta_{eh} * \left[ I_{i=1} * \psi_h + I_{i=0} * (1 - \psi_h) \right] * E_{hdp}(t) + \delta_f * I_{hdip}(t)$$

$$- A_{hdip}(t) *$$

$$\left[ \begin{array}{l} \mu_{bl} + I_{h=0} * \left( \mu_{ti} + v_i + \sigma_{0dip} + \kappa_{0dip} + I_{d=0} * \alpha si_{0ip} + I_{d=0} * \alpha sm_{0ip} + I_{d=1} * \alpha im_{0ip} \right) \\ + I_{h=1} * \left( \mu_h + \mu_{th} + \sigma_{1dip} + \kappa_{1dip} + I_{d=0} * \alpha si_{1ip} + I_{d=0} * \alpha sm_{1ip} + I_{d=1} * \alpha im_{1ip} \right) \end{array} \right]$$

$$- I_{h=0} * \left[ \theta * A_{0dip}(t) \right] + I_{h=1} * \left[ \theta * A_{0dip}(t) \right] \tag{4}$$

where $\delta_{eh}$ is the rate of progression from early active TB, $I_{eq}$ is an indicator function (= 1 if the condition $eq$ is met, 0 otherwise), $\psi_h$ is the proportion of active TB that is highly infectious (smear-positive), $\delta_f$ is the rate (1/duration) of ineffective therapy, $\mu_{bl}$ is the baseline mortality rate, $\mu_{ti}$ is the TB-specific mortality rate for non-HIV-associated TB, $\mu_h$ is the HIV-specific mortality rate, $\mu_{th}$ is the TB-specific mortality rate for people living with HIV, $v_i$ is the spontaneous recovery rate, $\sigma_{hdip}$ is the rate of diagnosis ultimately leading to successful treatment, $\kappa_{hdip}$ is the rate of placing individuals on ineffective treatment that does not generate acquired resistance, $\alpha si_{hip}$ is the rate of placing individuals on ineffective treatment that generates INH monoresistance, $\alpha sm_{hip}$ and $\alpha im_{hip}$ are the rates of placing individuals on ineffective treatment that generates MDR-TB, and $\theta$ is the HIV incidence rate. Thus, individuals enter this compartment through progression from early active disease or relapse/failure after ineffective treatment. They exit through spontaneous recovery, diagnosis leading to successful treatment, initiation of ineffective treatment (which can, in turn, generate acquired drug resistance), or death, and they acquire HIV infection at a constant rate.

## Active 'pre-treatment' compartment (*P*)

This compartment consists of individuals who have initiated a diagnostic attempt that will lead to successful treatment, yet remain infectious while awaiting initiation of treatment. Inclusion of this compartment is designed to capture the effects of diagnostic delays.

$$\frac{dP_{hdip}(t)}{dt} = \sigma_{hdip} * A_{hdip}(t) - P_{hdip}(t) * \left[ \begin{array}{l} \mu_{bl} + I_{h=0} * \left( \mu_{ti} + v_i + \rho_{0dip} \right) \\ + I_{h=1} * \left( \mu_h + \mu_{th} + \rho_{1dip} \right) \end{array} \right]$$

$$- I_{h=0} * \left[ \theta * P_{0dip}(t) \right] + I_{h=1} * \left[ \theta * P_{0dip}(t) \right] \tag{5}$$

where $\sigma_{hdip}$ is the rate of initiating successful diagnostic attempts, $I_{h=0}$ is an indicator function of HIV status (= 1 if $h = 0$, 0 otherwise), $\mu_{bl}$ is the non-TB mortality rate, $\mu_{ti}$ is the TB-specific mortality rate for non-HIV-associated TB, $\mu_h$ is the HIV-specific mortality rate, $\mu_{th}$ is the TB-specific mortality rate for people living with HIV, $v_i$ is the spontaneous recovery rate, $\rho_{hdip}$ is the rate of starting therapy after initiating a successful diagnostic attempt (1/diagnostic delay), and $\theta$ is the HIV incidence rate. Thus, individuals enter this compartment by initiation of successful diagnostic attempts and exit through initiation of treatment, spontaneous cure, or death. They acquire HIV infection at a constant rate.

## Active 'inappropriately treated' compartment (*I*)

This compartment contains all individuals being treated whose treatment course ends in default or failure. Unlike the previous (successful treatment) compartment, diagnostic delay is not explicitly

incorporated into this compartment, as to do so would prolong the duration of time until individuals who default re-enter the 'late active' compartment. Inclusion of a diagnostic delay before this compartment does not materially affect results. Individuals exit this compartment either in default after partial therapy—which has a defined probability of achieving cure despite not being completed—or failure. Failure leads immediately to another course of treatment, which can either be successful (i.e., transition to the latent compartment) or unsuccessful (i.e., remain in the inappropriate treatment compartment—which results in transition to the 'previously treated' compartment, p=1, not shown below).

$$
\begin{aligned}
\frac{dI_{hdip}(t)}{dt} &= \kappa_{hdip} * A_{hdip}(t) \\
&+ I_{d=1} * \alpha si_{hdip} * A_{h0ip}(t) + I_{d=2} * \left[ \alpha sm_{hip} * A_{h0ip}(t) + \alpha im_{hip} * A_{h1ip}(t) \right] \\
&- I_{hdip}(t) * \left[ \begin{matrix} \mu_{bl} + I_{h=0} * \left( \mu_{t0} + v_i + \delta_f \right) \\ + I_{h=1} * \left( \mu_h + \mu_{th} + \delta_f \right) \end{matrix} \right] \\
&- I_{h=0} * \left[ \theta * I_{0dip}(t) \right] + I_{h=1} * \left[ \theta * I_{0dip}(t) \right]
\end{aligned}
\tag{6}
$$

where $\kappa_{hdip}$ is the rate of placing individuals on ineffective treatment that does not generate acquired resistance, $\alpha si_{hip}$ is the rate of placing individuals on ineffective treatment that generates INH monoresistance from drug-susceptible TB, $\alpha sm_{hip}$ is the rate of placing individuals on ineffective treatment that generates MDR-TB from drug-susceptible TB, $\alpha im_{hip}$ is the rate of placing individuals on ineffective treatment that generates MDR-TB from INH-monoresistant TB, $\mu_{bl}$ is the non-TB mortality rate, $\mu_{ti}$ is the TB-specific mortality rate for non-HIV-associated TB, $\mu_h$ is the HIV-specific mortality rate, $\mu_{th}$ is the TB-specific mortality rate for people living with HIV, $v_i$ is the spontaneous recovery rate, $\delta_f$ is the rate (1/duration) of ineffective therapy, and $\theta$ is the HIV incidence rate. Thus, individuals enter this compartment through ineffective treatment from the late active compartment (conditional on whether that treatment also generates new drug resistance) and exit through completion of a course of ineffective therapy, spontaneous resolution, or death. They acquire HIV infection at a constant rate.

## Model fitting

The equations above comprise a series of 100 ordinary differential equations. In order to generate an equilibrium population according to user specifications of TB incidence, MDR-TB prevalence, and HIV prevalence, it is necessary to solve for the roots of a system of these 100 equations, plus three additional equations to account for the user inputs. We accomplish this using the 'fsolve' routine in SciPy (www.scipy.org). To solve for these three additional equations, we first match each user input to a single parameter: TB incidence to the transmission rate $\beta$, MDR-TB prevalence to the relative reduction in transmission for MDR-TB $\varphi_2$, and HIV prevalence to the HIV incidence rate $\theta$. We then constrain the system of equations such that the total population remains constant at 100,000, and we add the following three equations to the system:

$$d\beta/dt = \left( \text{calculated TB incidence} \right) - \left( \text{user-defined TB incidence} \right)$$

$$d\varphi_2/dt = \left( \text{calculated MDR-TB prevalence} \right) - \left( \text{user-defined MDR-TB prevalence} \right)$$

$$d\theta/dt = \left( \text{calculated HIV prevalence} \right) - \left( \text{user-defined HIV prevalence} \right)$$

By solving for the roots at which this system of equations equals zero, we generate an equilibrium (steady-state) population that also defines $\beta$, $\varphi_2$, and $\theta$ such that the user-defined targets are also met.

Additional complexity is added to account for non-equilibrium in TB incidence and MDR-TB prevalence. We accomplish this by fitting an equilibrium defined by the user-specified target of HIV prevalence, and the user-specified 'initial' values of TB incidence and MDR-TB prevalence. This equilibrium is set to be 50 years in the past; at this time, the system of equations is solved as above. We then allow for the parameters $\beta$ and $\varphi_2$ to be altered from their original values (corresponding to the equilibrium condition) such that a second set of targets are attained after 50 years (start of the analysis). If these

calculated values at the end of 50 years do not match the user-defined targets, the parameter values are changed accordingly, and the model is re-run from equilibrium until the appropriate parameter value is identified that generates the user-defined TB incidence and MDR-TB prevalence, within a relative tolerance of 0.05 in each variable. These parameters may then be further modified such that they change over the analysis frame of five years (e.g., to describe an epidemic with increasing MDR-TB prevalence over time).

In the primary version of the model, this is only automated for low-incidence scenarios in which the user inputs a TB incidence of less than 50 per 100,000/year. In such cases, the model generates an equilibrium population 50 years in the past with a TB incidence of 50 per 100,000/year and reduces the transmission parameter $\beta$ until the user-defined TB incidence is achieved 50 years later (i.e., the start of the analytic period). This accounts for the fact that most low-incidence settings have substantially more latent TB infection than would be estimated by an equilibrium population with a very low TB incidence—thereby more appropriately reflecting a higher proportion of TB due to reactivation rather than recent infection. However, we include code (described in the user manual below) that allows users to define high incidence scenarios that are likewise not at equilibrium, as well as 'emerging MDR' scenarios in which the prevalence of MDR-TB is increasing rather than stable.

## Acknowledgements

We are grateful to Jeff Pennington for development of the web interface, and to Adithya Cattamanchi, Madhukar Pai, Jonathan Golub, and Richard Chaisson for helpful comments.

## Additional information

### Funding

| Funder | Grant reference number | Author |
|---|---|---|
| National Institutes of Health (NIH) | R21AI101152 | David W Dowdy |
| Canadian Institutes of Health Research | MOP 271997 | David W Dowdy |

The funder had no role in study design, data collection and interpretation, or the decision to submit the work for publication.

### Author contributions

DWD, Conception and design, Acquisition of data, Analysis and interpretation of data, Drafting or revising the article; JRA, PJD, RHG, Conception and design, Analysis and interpretation of data, Drafting or revising the article

## Additional files

### Supplementary files
• Supplementary file 1. Model code

• Supplementary file 2. Model interface and user manual

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
