## [Decision Letter]

Thank you for sending your work entitled “A user-friendly, open-source tool to project impact and cost of diagnostic tests for tuberculosis” for consideration at *eLife.* Your article has been favorably evaluated by Prabhat Jha (Senior editor) and 2 reviewers, one of whom is a member of our Board of Reviewing Editors.

The Reviewing editor and the other reviewer discussed their comments before we reached this decision, and the Reviewing editor has assembled the following comments to help you prepare a revised submission.

The group has developed a dynamic transmission model of TB that projects impact (TB incidence, TB mortality, MDR TB prevalence) and incremental cost of scaling up nine TB diagnostic interventions customized to local conditions. Two features that differentiates it from other TB models are: 1) its ability to be customized to local conditions using input specifications of TB incidence/MDBR TB prevalence/HIV prevalence/costs) and 2) being a simplified model freely available on a web-based platform that can be manipulated by non-experts to make programmatic decisions. The authors compared the nine diagnostic strategies across four other epidemiological settings.

They demonstrate that regardless of the scenario, the interventions rank in the same order in terms of their impact and overall cost. However, actual costs and impact may differ.

The field needs these simplified tools and needs to empower local decision makers to make choices that are relevant to their context. We do however have comments and would appreciate if the authors could address these:

General comments:

1) The provision of a tool does not mean that the tool will be used and it is often the case that it does not. In country dissemination and training is needed. The use is strengthened if a global body (e.g. WHO) advocates for this. Have the authors tested/disseminated this tool with global and country level stakeholders? What is their plan to make sure that this tool does get used?

2) It would be great if the authors could comment on other simplified tools that are available for TB or HIV and how these tools have transformed the field (if at all).

3) Is the availability of data a limiting factor for these models to be used at local a local level (national or sub-national). A key input parameter to the model is TB incidence. How available is localized TB incidence data? At national or sub-national data? Similar comment regarding intervention costs.

4) The model assumes that everything else remains constant overtime (e.g. TB incidence). How does this affect the results? Is this the reason that projections are only 5 years out.

While the authors do discuss the limitations, I think a greater discussion on the limitation of the model is needed in the discussion. This should include, where relevant, greater discussion of sensitivity analyses.

Results section:

5) It would be good to have confidence intervals for the model outcomes for the high incidence scenario (global validation).

6) Sensitivity analysis: I would be interested to see some sensitivity analysis around unit cost given that this is one of the most difficult data points for programs to have especially when they are considering scaling up new interventions. Please comment on this more.

---

## [Author Response]

General comments:

*1) The provision of a tool does not mean that the tool will be used and it is often the case that it does not. In country dissemination and training is needed. The use is strengthened if a global body (e.g. WHO) advocates for this*. *Have the authors tested/disseminated this tool with global and country level stakeholders? What is their plan to make sure that this tool does get used?*

We agree wholeheartedly with this comment. In response, we have added the following as the fourth paragraph of the Discussion:

“For any such simplified transmission model to have impact on decision-making, it is important to consider who the eventual users of such a model might be, and to develop the model in consultation with those groups. In this case, we identified a number of potential end-user organizations including technical assistance agencies (e.g., KNCV Tuberculosis Foundation), funding bodies (e.g., Global Fund for AIDS, Tuberculosis, and Malaria), non-governmental organizations (e.g., Medecins Sans Frontiers), National Tuberculosis Programs, and regional offices of the World Health Organization. We then invited representatives of these organizations to attend a one-day workshop (April 2014, The Hague, The Netherlands) describing methods and challenges related to modeling of TB diagnostics in general, including this transmission model in particular. We developed “hands-on” exercises for participants to better learn the model and solicited specific feedback, which is being incorporated into the model structure and web interface in ongoing fashion. We next intend to develop a series of informal “case studies” whereby the use of this model for in-country decision-making can be demonstrated and disseminated.”

*2) It would be great if the authors could comment on other simplified tools that are available for TB or HIV and how these tools have transformed the field (if at all)*.

As also commented in the first minor comment below, the obvious example of a simplified tool that has been used for decision-making in HIV is the Spectrum suite of models. We now include the following text in our Introduction:

“Simplified models designed for in-country use by decision-makers, most notably the Spectrum suite of models supported by the Futures Institute, have been used to inform decision-making in the fields of reproductive health and human immunodeficiency virus (HIV) for over a decade. Estimates from the Spectrum models are now routinely incorporated into official global and country-level estimates of HIV disease burden and intervention impact. Other simplified models are readily available for impact projections related to non-infectious diseases, where transmission assumptions are less important. However, to date, simple, user-friendly transmission models have not been widely used for decision-making related to many infectious diseases other than HIV.”

*3) Is the availability of data a limiting factor for these models to be used at local a local level (national or sub-national). A key input parameter to the model is TB incidence. How available is localized TB incidence data? At national or sub-national data? Similar comment regarding intervention costs*.

We agree that data availability may be a limiting factor, but we also believe that local decision-makers are routinely forced to make decisions in the absence of such data. This model may still be useful in such circumstances by (a) allowing users to see the effects of different assumptions (e.g., different TB incidence or unit costs) in real-time, and (**B**) enabling users to link such assumptions to best available data on other parameters (e.g., TB natural history or epidemiology) that may have more data to inform them. The alternative, of course, is to disregard all such data and leave local decision-making to a non-transparent process of “expert opinion.” In response to this comment, we have included the following in our Limitations section:

“Data to inform even these three parameters – as well as unit costs – are unlikely to be available on sub-national levels. As a result, users wishing to adapt the model to a smaller geographic scale will need to perform additional data-gathering exercises to inform these estimates if they wish to maximize the utility of the model.

Nevertheless, even if high-quality data are not available at the local level, this model allows decision-makers to estimate epidemiological and economic values according to reasonable assumptions (e.g., comparison of TB notification rates or budget line-items to those in the published literature or at the national level), vary those assumptions in real-time, and obtain corresponding projections of comparative impact that incorporate best available current data on epidemiological or natural history parameters (e.g., TB progression and reactivation).”

*4) The model assumes that everything else remains constant overtime (e.g. TB incidence). How does this affect the results? Is this the reason that projections are only 5 years out*.

This is, indeed, one of the primary reasons for limiting projections to five years, over which time the majority of TB epidemics are reasonably stable. We note, however, that our low TB incidence settings do not assume constant TB incidence over time, and that users can adapt the model to incorporate declining (or rising) TB incidence at baseline. In general, incorporating declining incidence assumptions leads to higher estimates of reactivation, which result in lower estimates of impact, as covered in our new sensitivity analyses (see below).

*While the authors do discuss the limitations, I think a greater discussion on the limitation of the model is needed in the discussion. This should include, where relevant, greater discussion of sensitivity analyses*.

In response to the comments above we have expanded the discussion of model limitations. We have also, as below, expanded our results and discussion of sensitivity analyses in the corresponding portions of the manuscript.

Results section:

*5) It would be good to have confidence intervals for the model outcomes for the high incidence scenario (global validation)*.

Thank you for this suggestion. We now provide 95% uncertainty ranges for the primary validation parameters reported, as well as methods for generating those ranges. In addition, we have used those same methods to generate 95% uncertainty ranges around each of the intervention estimates in each country’s baseline scenario – these ranges are now visible as “crosshairs” in the corresponding scatterplot figures on the website, and they also appear in the web-based comparison table.

*6) Sensitivity analysis: I would be interested to see some sensitivity analysis around unit cost given that this is one of the most difficult data points for programs to have especially when they are considering scaling up new interventions. Please comment on this more*.

We have added additional methods and results describing sensitivity analyses on cost, as well as key natural history parameters. In the latter, we describe the importance of three natural history parameters: empiric treatment proportion, pre-diagnostic delay, and the reactivation rate. Of note, we have now also added “radio buttons” on the website scatterplot to demonstrate the impact of changing these three parameters, for rapid assessment of their impact on the website.